# Predictors of Suicide Risk and Mental Health Outcomes among Hong Kong Veterinarians: A Cross-Sectional Study

**DOI:** 10.3390/bs13090770

**Published:** 2023-09-15

**Authors:** Camille K. Y. Chan, Paul W. C. Wong

**Affiliations:** Department of Social Work and Social Administration, The University of Hong Kong, Hong Kong, China

**Keywords:** burnout, compassion satisfaction, compassion fatigue, mental health, suicide, depression, anxiety, veterinarians, predictors

## Abstract

The professional quality of life (ProQOL) in the veterinary profession has gained increasing attention, yet little is known about its association with the mental health status of Hong Kong veterinarians. This study aimed to examine the impact of elements that make up ProQOL on the risk of suicide, depression, and anxiety among Hong Kong veterinarians. All veterinarians registered and practicing in Hong Kong at the time of recruitment were eligible to take part in the e-survey study between 1 January and 31 March 2022. Binary logistic regression was used to analyze the data from 56 participants. The results showed that 22.0% of the participants were at risk of suicide, 19.6% had current suicidal ideation, 29.4% had depression, and 29.4% had anxiety symptoms. The findings suggested poor mental health status among Hong Kong veterinarians and were comparable to or more prevalent than those reported in Anglophone and European countries. Results from the binary logistic regression suggested that burnout was a predictor of depressive symptoms, and that secondary traumatic stress showed potential in predicting suicide risk. Our study, however, did not find conclusive evidence supporting compassion satisfaction as a predictor of reduced symptoms of common mental issues. Further investigation into contextual factors affecting the mental health of veterinarians in Hong Kong is warranted. Improving the profession’s mental health literacy and self-efficacy should be prioritized as a suicide prevention strategy to enhance mental health awareness.

## 1. Introduction

The suicidality and poor mental health of veterinary professionals have drawn increasing attention. Various studies have indicated that veterinarians experience high levels of depression (ranging from 6% to 28%) [1,2,3,4,5] and anxiety (ranging between 4% and 33%) [1,2,3,4]. The suicide rate of veterinarians in the U.S. is higher than that of the general U.S. population [6]. Previous research suggested the complexity of veterinarians’ roles such as long working hours, heavy caseloads, engagement in euthanasia [1,7,8], the challenging nature of balancing clients’ expectations and medical interests [9], poor workplace culture [10], and isolated working conditions are predictors of veterinarians’ poor mental health status. Furthermore, studies have found that veterinarians with poor mental health are likely to experience burnout, higher levels of secondary traumatic stress, and lower compassion satisfaction [2,4,11,12,13,14]. Studies have also highlighted the easier access to pentobarbital [15], a drug commonly used for euthanasia of animals, to be a significant factor in their suicidal behavior. Little is known about the prevalence of the mental health status and professional quality of life (ProQOL) of Hong Kong veterinarians. 

It is important to note that contextual factors in Hong Kong can play an important role in affecting the mental health situations of the local veterinarians. Being one of the most urbanized cities in the world, Hong Kong faces high rental costs and a competitive business environment that adds significant stressors to the veterinary profession [16]. For instance, unlike many other countries and cities, investing in veterinary clinics in Hong Kong is not restricted to licensed veterinarians; hence, the relatively low market entry barrier possibly amplifies rivalry among veterinary clinics [17].

This quantitative study aims to examine the mental health status of veterinarians in Hong Kong, specifically focusing on symptoms of some common mental health issues such as depression, anxiety, suicide risk, and elements of the professional quality of life including burnout, secondary traumatic stress, and compassion satisfaction. 

This research seeks to explore the presence of these conditions among Hong Kong veterinarians and investigate whether ProQOL impacts their mental health. We hypothesized that (i) Hong Kong veterinarians have high levels of poor mental health, (ii) higher levels of compassion satisfaction is associated with fewer symptoms of depression and anxiety and lower suicide risk among Hong Kong veterinarians, and (iii) higher levels of burnout or secondary traumatic stress are associated with more symptoms of depression and anxiety and an increased level of suicide risk.

## 2. Methods

### 2.1. Study Design

This cross-sectional study investigated how the presence of burnout, secondary traumatic stress, and compassion satisfaction among veterinarians practicing in Hong Kong affect their levels of suicidality, depression, and anxiety. Potential respondents were guided to a self-administered e-survey that included validated tools to assess these predictors and outcomes of interest. 

### 2.2. Setting

All Hong Kong veterinarians from this study participated in an online survey between 1 January 2022 and 31 March 2022. Veterinarians were invited by an e-invitation distributed by (i) the Hong Kong Veterinary Association, to approximately 230 members; (ii) the City University of Hong Kong’s College of Veterinary Medicine and Life Sciences, to approximately 40 veterinarian colleagues; (iii) the research team’s personal network, to 30 veterinarians, nurses, and animal volunteers who worked closely with veterinarians; (iv) a Facebook group exclusive to Hong Kong veterinarians consisting of approximately 200 members; and (v) referrals from research participants. To reach as many potential respondents as possible, at least one reminder was sent from each channel about two months after the first invitation was sent.

### 2.3. Participants

Veterinarians who were registered in the Veterinary Surgeon Board of Hong Kong, were practicing in Hong Kong, and were not retired at the time of recruitment were eligible to take part in this study. Interested potential participants were asked to click an online link or scan a QR code [18] and were directed to an information sheet about this research. They provided their consent by checking a box, indicating their agreement with the information sheet. Then, they were invited to complete the e-screening questions to assess their eligibility for this study. Those who passed the e-screening were guided to a self-administered e-survey. Due to Hong Kong’s bilingual environment where citizens are typically exposed to English from an early age and possess proficiency in the language and considering that all veterinarians were trained overseas mainly in English speaking countries, all research materials, including the invitation, information sheet, screening questions, and e-survey were presented in the English language rather than Traditional Chinese.

### 2.4. Variables

Information about participants’ demographic and profession-related information, such as the year of their earliest qualification, medical specialty, employment status, work position, type of workplace, and hours of work in an average week were collected. We measured veterinarians’ professional quality of life with the ProQOL scale [19] and their mental health status with the Suicidal Behaviors Questionnaire-Revised (SBQ-R) [20], the PHQ-9 [21], and the General Anxiety Distress-7 (GAD-7) [22]. We also examined other aspects that were related to veterinarians’ wellbeing but they were not included in the analyses of this study, such as occupation demand and resources, self-compassion, euthanasia distress, fearlessness of death, experience of cyberbullying, etc. The questionnaire required about 20–25 min for completion and can be found in the Appendix A.

### 2.5. Measurement

#### 2.5.1. Professional Quality of Life Scale (ProQOL)

The ProQOL scale [19] is a self-reported questionnaire that measures compassion fatigue and compassion satisfaction of helping professions. The 30 items contain 3 sub-scales that assess burnout, secondary traumatic stress, and compassion satisfaction, and are measured on a 5-point Likert scale. The scores are calculated by adding up each sub-scale after recording the 5 reversed items of the burnout sub-scale. The sub-scale scores that ranged from 10 to 50 can be categorized as low (22 or below), moderate (23 to 41), or high (42 or above). The Cronbach α values for burnout, secondary traumatic stress, and compassion satisfaction sub-scales were α = 0.79, 0.87, and 0.93, respectively, indicating good internal consistency and reliability.

#### 2.5.2. Suicidality

The Suicidal Behaviors Questionnaire-Revised (SBQ-R) [20] consists of four items that assess aspects of suicide-related thoughts and behaviors, including lifetime suicidal ideation or attempt; the frequency of suicidal ideation over the last 12 months; the threat of suicide attempts; and the likelihood of future suicidal behaviors. The SBQ-R total score ranged from 3 to 18, and a score of 7 or above indicates being at risk of suicide. The Cronbach α for the four items scale was α = 0.85, which indicated good internal reliability of the scale. 

Item number 9 of the Patient Health Questionnaire-9 (PHQ-9) [21] was also used to measure the current risk of participants’ suicidal ideation in the past two weeks.

#### 2.5.3. Depressive Symptoms

The PHQ-9 [21] is a 9-item multipurpose instrument for screening, diagnosing, monitoring and measuring the severity of depression [23]. Each item is assessed on a scale from 0 (not at all) to 3 (nearly every day), adding up to 0–27, whereby higher PHQ-9 scores indicate greater depressive symptoms. The symptom severity can be categorized as none to minimal (0–4 points), mild (5–9 points), moderate (10–14 points), moderately-severe (15–19 points), and severe (20–27 points). PHQ-9 is a reliable scale with 88% sensitivity and specificity. The Cronbach α was 0.86, indicating excellent internal reliability of the scale [21].

#### 2.5.4. Anxiety Symptoms

The General Anxiety Distress-7 (GAD-7) [22] is a 7-item multifunctional instrument for screening, diagnosing, and assessing the severity of anxiety [23]. Each item is scored from 0 (not at all) to 3 (nearly every day), with 21 as the highest possible score. The severity of anxiety symptoms can be categorized as none to minimal (0–4 points), mild (5–9 points), moderate (10–14 points), and severe (15–21 points). GAD-7 is a reliable and validated anxiety survey with a sensitivity of 89% and specificity of 82% [24], and the Cronbach α was 0.92, indicating excellent internal reliability of the scale.

### 2.6. Bias

Our online survey sought to alleviate self-selection and non-response biases by implementing a multi-channel approach for survey dissemination. This comprehensive dissemination strategy targeted a diverse range of veterinarians through various platforms, including professional and educational organizations, social media, as well as personal networks, ensuring that the survey reached its intended participants and minimizing self-selection bias. Additionally, reminders were sent through these channels to encourage participation and reduce non-response bias.

### 2.7. Study Size

The sample size for this study was determined by considering the statistical analysis methods. A sample size of at least 30 participants was set for each of the 3 sets of binary logistic regression planned, considering the general rule of thumb that recommends having 10 to 20 participants per predictor variable [25,26]. To further strengthen the statistical power and reliability of the findings, our goal was to recruit a total of 60 participants for this study. 

### 2.8. Statistical Methods

Descriptive statistics, including means, frequencies, and percentages, were used to describe the sample. To determine the prevalence of the risk of suicide, depression, and anxiety, we calculated the proportion of respondents with SBQ-R scores ≥ 7 for at risk of suicide; PHQ-9 scores > 9 for depression; and GAD-7 scores > 9 for anxiety. The ninth item of the PHQ-9 > 0 indicated respondents with current suicidal ideation. The assessment of the ProQOL sub-scales involved calculating the proportion of respondents with sub-scale scores ≥ 42. Binary logistic regression analyses were conducted to assess the relationship between veterinarians’ ProQOL and mental health outcomes. This approach allowed us to explore the specific associations between predictor variables, namely the raw scores of burnouts, secondary traumatic stress, compassion satisfaction of ProQOL sub-scales, and binary outcomes relating to veterinarians’ risk of suicide, current suicidal ideation, depression, and anxiety. We found this test to be well-suited for our analysis, considering the limited sample size and non-normal distribution observed in our outcome variables.

The strength and direction of the linear relationship between hours of work per week and the SBQ-R, PHQ-9, and GAD-7 scores were explored using correlation analyses. Each scale or sub-scale’s reliability was assessed using Cronbach’s alpha; values above 0.7 indicates good reliability in this population. Missing values involving scores calculation were excluded listwise. Therefore, participants who did not answer all the questions within each set of the ProQOL, SBQ-R, PHQ-9, or GAD-7, with data relating to that score, were not considered in the calculations and were excluded from the analysis. Statistical analyses were conducted using SPSS for Windows v. 28 [27]. All significant tests were two-tailed, and findings with a *p*-value of 0.05 were considered statistically significant.

This study’s protocol was reviewed and approved by the Human Research Ethics Committee at the University of Hong Kong (reference number: EA200192).

## 3. Results

### 3.1. Participants and Descriptive Data

Sixty-five respondents took part in the e-survey. The response rate accounted for 6.6% of registered veterinarians in Hong Kong [28]. Those who had completed less than 15.0% of the questionnaire (*n* = 9) were excluded from the analysis. Hence, data from 56 participants, with a 90.7% data completion rate (see Appendix A), were included in the analysis. Due to this study’s inability to capture information regarding how respondents became aware of the survey, as well as lacking knowledge of the number of potential participants reached through each survey distribution channel, we were unable to estimate the response rate by channel.

Table 1 provides an overview of the respondents’ background information. The average hours of work per week were 46.6 ± 8.8 for respondents who worked full time (*n* = 43) and 27.8 ± 11.1 for those who worked part time (*n* = 13). Pearson’s correlation identified a weak positive relationship between average hours of work per week with GAD-7 scores (r = 0.28, *p* < 0.046); a moderate positive relationship between SBQ-R with PHQ-9 (r = 0.46, *p* < 0.001) and with GAD-7 (r = 0.37, *p* < 0.001); and a strong positive relationship between PHQ-9 and GAD-7 (r = 0.82, *p* < 0.001).

### 3.2. Risk of Suicide, Suicidal Ideation, and Behaviors

The SBQ-R mean score was 5.8 ± 3.4 and indicated 22.0% (*n* = 11) were as at risk of suicide. It was found that 50.0% (*n* = 25) of the participants never had suicidal ideation in their lifetime; 34% had suicidal ideation; 14.0% had plans for suicide; and 2.0% had attempted suicide. Regarding suicidal thoughts and behaviors in the past year, 66.0% (*n* = 33) of the participants had never thought of killing themselves; 16.0% had thought of it once; 8.0% considered it twice; 4.0% had thought of killing themselves three to four times; and 6.0% had thought of it more than four times in the past year. Item nine of the PHQ-9 identified 19.6% (*n* = 10) had thoughts they would be better off dead or hurting themselves in the past two weeks. 

Out of all participants who have conceived killing themselves at some stage of their lives (50%, *n* = 25), 32.0% have not told anyone that they might attempt suicide; 8.0% disclosed it once; and 10.0% disclosed it more than once. While 66.0% (*n* = 33) claimed there was no chance that they would attempt suicide in the future, 18% indicated that it would be rather unlikely; 14% stated it was likely; and 2% reported that it was very likely.

### 3.3. Depression and Anxiety: Symptom Severity and Presence

None of our respondents reported having severe depressive symptoms; 7.8% indicated evidence of moderately severe depressive symptoms; and 21.6% had moderate depressive symptoms. Most respondents had minimal (41.2%) or mild symptoms (29.4%) of depression. The overall PHQ-9 mean score ranged between 0 and 19, with an average score of 6.8 ± 5.3 out of a possible 27 points.

Anxiety was prevalent among 29.4% (*n* = 15) of participants. A total of 19.6% and 9.8% of participants had moderate and severe anxiety symptoms, respectively. Almost half of the respondents had minimal anxiety symptoms (47.1%) and 23.5% had mild symptoms. The GAD-7 mean score also ranged between 0 and 19, with an average score of 6.3 ± 5.4 out of a possible 21 points for anxiety.

Out of the 15 respondents with anxiety or depression, 12 were classified as comorbid with both conditions, representing 23.5% of respondents (see Appendix A). The Chi-square test of independence showed significant relationships between current suicidal ideation and depression (χ^2^ = 5.61, df = 1, *p* = 0.018); anxiety (χ^2^ = 5.61, df = 1, *p* = 0.018); and comorbidity (χ^2^ = 9.20, df = 1, *p* = 0.002). Significant relationship between participants who were at risk of suicide and anxiety (χ^2^ = 4.05, df = 1, *p* = 0.044) were also confirmed using the Chi-square test.

### 3.4. ProQOL among Hong Kong Veterinarians

When examining the distribution of burnout, secondary traumatic stress, and compassion satisfaction among our study participants (Appendix A), we found that 23.2% of participants reported low levels of burnout; while 73.2% reported moderate levels; and only 3.6% reported high levels of burnout. In terms of secondary traumatic stress, we observed that 33.9% of participants reported low levels, while the majority, comprising 66.1%, reported moderate levels. Interestingly, none of the participants reported high levels of secondary traumatic stress. Regarding compassion satisfaction, we found that 3.6% of participants were identified with low levels; a significant majority of 76.8% reported moderate levels; and the remaining 19.6% indicated high levels of compassion satisfaction.

### 3.5. ProQOL’s Association to Mental Health Outcomes

Binary logistic regression was performed to examine the predictive relationship between the ProQOL sub-scales with various mental health outcomes, including (i) being at risk of suicide; (ii) having current suicide ideation; (iii) depression; and (iv) anxiety (see Table 2). In terms of suicidality, the Nagelkerke R-square suggested that the ProQOL sub-scale accounted for 33.9% of the variance in being at risk of suicide. Among the sub-scale, secondary traumatic stress emerged as a significant predictor of being at risk of suicide (*p* < 0.05, CI: 1.01–1.55). A one-unit increase in secondary traumatic stress was associated with a 1.251 times higher risk of being at risk of suicide. Compassion satisfaction showed a marginal effect on reducing the risk of suicide (*p* = 0.089), indicating that every unit increase in compassion satisfaction was associated with a 15.2% decreased in the risk of being at risk of suicide. For current suicide ideation, although the Nagelkerke R-square indicated the ProQOL sub-scale explained 44.1% of the variance, none of the sub-scales were found to be a significant contributor to the model (*p* > 0.01).

The analysis found the ProQOL sub-scales accounted for 58.8% of the variance in depression, with burnout emerging as a significant predictor (*p* < 0.05; CI: 1.01–1.64). Each unit increase in burnout corresponded to a 28.5% higher risk of having depression. The association between secondary traumatic stress and depression was marginally significant (*p* = 0.064), suggesting that every unit increase in secondary traumatic stress was associated with a 16.6% higher risk of depression.

Furthermore, the Nagelkerke R-square suggested that the ProQOL sub-scale explained 48.4% of the variance in anxiety. However, both the burnout and secondary traumatic stress scores were marginally outside the 95% confidence interval (*p* = 0.069 and 0.088, respectively). It was found that each unit increase in burnout and secondary traumatic stress scores corresponded to a 22.1% and 13.9% higher risk of anxiety, respectively.

## 4. Discussion

This first-of-its-kind quantitative study explored the mental health landscape and the ProQOL (as well as their interaction) of veterinarians in Hong Kong. Our respondents reported a lower risk of suicide than is estimated in the existing literature (Australia at 26.0–29.6% [7] and Germany at 32.1% [5]). While this study did not investigate the reasons behind the relatively low risk of suicide, some veterinarians explained that workplace harmony and clients’ satisfaction can be rewarding, which may have acted as a protective factor in preventing suicidal behavior [17].

Our hypothesis posited that Hong Kong veterinarians exhibit poor mental health status. This study found high levels of depressive symptoms among Hong Kong veterinarians (29.4%) and is comparable to or higher than those in Germany (28.8%) [5], Australia (19.8%) [3], Canada (8.9–9.0%) [2,4], and the U.K. (5.8%) [1]; oppositely, anxiety of Hong Kong veterinarians (29.4%) is comparable to those in the U.K. (26.3%) [1] and Canada (23.6–29.0%) [2,4], but higher than in Australia [3]. We also found comorbidity of depression and anxiety were common among Hong Kong veterinarians (23.5%), which is higher than of those in Australia (4.5%) and Canada (7.1–7.2%) [2,4]. The prevalence of suicidal ideation of Hong Kong veterinarians in the past 12 months (19.6%) is similar to those in Germany (19.2%) [5].

Despite common sources of stress, such as workplace demand and client expectations, which are shared among the profession, a study suggested that contextual factors in Hong Kong contributed to unique stressors that were uncommon in Anglophone and European countries [17]. Being one of the world’s most populous cities, Hong Kong veterinarians reported being most stressed about sustaining their business under extremely high rental costs and intense competition [16,17]. The profession’s stress was further heightened by its geographic convenience due to clients’ price shopping and transfer between veterinary clinics [17]. Veterinarians worried about Hong Kong’s lack of business regulatory requirements for veterinary clinic ownership, as current legislation permits non-veterinarian owners to exclusively own medical practices, which could hinder the legal liability of non-veterinarian investors [17]. They are concerned that businessman-owned veterinary clinics are less familiar with veterinary ethics, such as the appropriate length of medical consultations and operating within an ethical profit model [17]. A previous study also found that workplace leadership and harmony have a role in veterinarians’ mental health [17]. 

The high levels of poor mental health status of veterinarians in Hong Kong are also comparable to that of the local health care professionals. A similar study that surveyed 393 Hong Kong physicians found that 16.0% were depressed, and 15.3% had suicidal ideation in the past 2 weeks [29]. Depression and anxiety were prevalent among 35.8% and 37.3% of Hong Kong nurses [30], respectively. In contrast, depression, anxiety, and ideation (within the past year) among the general population were 10.7%, 4.2%, and 1.2%, respectively [31,32]. Based on our findings, veterinarians were more prone to depression (29.4%) and suicidal ideation within the past 2 weeks (19.6%) than other health care professionals in Hong Kong. We believe the healthcare profession’s continuous engagement of keeping up with healthcare development contributed to additional stress in comparison to the general population. While we know little about veterinarians’ willingness to access mental health services, a study indicated the low priority to physician mental health is embedded in the culture of medicine where help-seeking is considered “punitive”, due to reasons such as discrimination in medical licensing, stigma related to the fitness to practice, and barriers to professional advancement [33,34].

This study has found a statistically significant, although weak, association between increased work hours and elevated levels of anxiety symptoms. Similar findings have been reported within the working population in Hong Kong, indicating that longer working hours are associated with a higher likelihood of experiencing anxiety symptoms [35]. This correlation becomes particularly noteworthy in the veterinary profession, given the nature of their work involving surgical procedures and demanding precision with little room for errors [2,17]. Although this study found that veterinarians working full time appeared to have higher levels of anxiety symptoms, the association did not reach statistical significance. Furthermore, previous research indicated that veterinarians’ main barriers to accessing mental health services in Hong Kong were due to their lack of work–life balance, resulting from their late finishing of work and unpredictable work schedule [17]. It is also worth noting that numerous previous studies have consistently reported significant associations between depression and anxiety, depression and suicidality, as well as anxiety and suicidality. These associations are also evident in the findings of this study.

We speculate that the veterinarian–client–patient dynamic poses additional challenges facing the veterinary profession, compared to physicians, and contributes to their poorer mental health. While veterinarians are trained to manage the health of their animal patients who cannot speak, they are anticipated to meet expectations of their human clients (usually the animals’ caretakers) who have a significant role in medical decision making on behalf of the animals [36]. This stress is further intensified by the intricate dynamics involved in managing client relationships, meeting their expectations, and navigating knowledge disparities that may arise [4,24]. Additionally, the repeated exposure to moral conflicts stemming from disagreements regarding treatment may lead to the development of compassion fatigue among veterinarians [3,7,9]. It has been reported that clients’ unrealistic expectations and complaints were significant stressors facing Hong Kong veterinarians [16]. Studies also suggested that, if these conflicts remain unresolved, clients may resort to expressing their grievances through social media channels, which can potentially result in veterinarians being publicly criticized [17]. 

However, the traditional curriculum of medical schools often focuses its training on biomedical and clinical skills, where medical professionalism comprising communication techniques, customers’ (or clients’) experience, workplace relations, and emotional wellbeing was neglected [37], despite its importance in preventing conflict and maintaining good mental health [17]. Although the Internet enabled convenient access of information on subjects such as animal welfare and caretaking of companion animals, research indicates that health misinformation is prevalent on social media [38], restricting animal caretakers’ aptitude to look after their pets. Discontented experiences and deepening distrust between clients and veterinarians can negatively affect the quality of veterinary medical services, their mental wellbeing, and, ultimately, the welfare of animals.

We hypothesized that higher levels of compassion satisfaction would be associated with fewer symptoms of depression and anxiety. However, our findings did not provide sufficient evidence to support compassion satisfaction as a predictor of reduced symptoms of depression and anxiety. Although we observed that higher compassion satisfaction was associated with a decreased risk of being at risk of suicide, its marginal statistical significance of this association prevents us from drawing a definitive conclusion. 

Our last hypothesis investigated the association between higher levels of burnout, secondary traumatic stress, more symptoms of depression and anxiety, as well as increased risk of suicide. We find partial support for this hypothesis, as secondary traumatic stress emerged as a significant predictor of being at risk of suicide, while burnout did not show statistical significance in predicting this outcome. Regarding depression, burnout was a significant predictor, with secondary traumatic stress nearly meeting the threshold for statistical significance. If we were to consider a slightly lower confidence level of 90%, secondary traumatic stress scores would also be deemed significant for depression. Similarly, for anxiety, both burnout and secondary traumatic stress scores would have been predictors if a 90% confidence interval was considered. However, when examining the prediction of current suicidal ideation, we found insufficient evidence to support the ProQOL as a predictor of this outcome.

The literature suggested compassion satisfaction could act as a protective factor of veterinarians’ distress [39,40]. It is a matter of concern to notice only one fifth of our participants reported high compassion satisfaction. We believe it is crucial to improve the profession’s compassion satisfaction, and this can be facilitated through organizational efforts of each clinic [40]. Interestingly, while the literature suggested veterinarians often experience high secondary traumatic stress [2,4,12,13], our findings suggested none of our respondents have identified with high secondary traumatic stress. It is worthwhile to delve into the cause, as the outcomes from the current field of knowledge are incongruous. We speculate that it could be associated with non-Western cultural dynamics and local circumstances in Hong Kong.

There are a few limitations to this research. First, our study represented only 6.6% of all Hong Kong veterinarians, which has a similar response rate to local mental health studies of the healthcare profession [30]. The small size of the profession restricted the generalizability of our findings and is susceptible to bias. Also, our questionnaire took around 20–25 min to complete, and that might contribute to the low response rate. Second, systematic missing values in gender disable sub-group analyses that address the gender paradox of suicide and depression [41]. Third, our research cannot dismiss the possible influence of COVID-19 on veterinarians’ mental wellbeing as data were collected during this pandemic. Last, we used multiple ways of universal recruitment to promote our survey to the potential study participants, but we did not ask how the participants found the survey in the questionnaire.

## 5. Implications

Since many practicing veterinarians may be too busy to seek professional help [17], we have recently piloted an animal-assisted mental health educational program with veterinary medical students in Hong Kong. This program aimed to prepare future veterinarians for possible stressors and challenges ahead, and was assisted by trained therapy dogs as evidence suggested animal-assisted education effectively engages mental health conversations [42] and reduces anxiety [43]. We also believe that because veterinarians have more love and care for animals than others; therefore, in the presence of animals, students are expected to have better socio-emotional engagement and adherence to memory [44]; thus, this encourages the learning of mental health literacy and sustainability when compared to non-animal-assisted programs. Improving veterinary professionalism, restoring veterinarian–client trust, and encouraging mental health assessments are crucial to the improvement of the veterinarians’ mental health and is beneficial to the welfare of animals, and “Not One More Vet” is necessitated to suicide.

## 6. Conclusions

Suicide and mental health disorders are a global concern affecting the public and high-risk professions such as veterinarians. Although the low response rate raised questions about the Hong Kong veterinarians’ willingness to explore and enhance their mental health, our findings suggested their mental illness was comparable to or more prevalent than those reported in Anglophone and European countries and is in need for more attention and efforts to improve this worrying situation. Future research should examine the profession’s coping strategies for stress, including the tendency to seek professional help and the usage of mental health resources. 

## Figures and Tables

**Table 1 behavsci-13-00770-t001:** Overview of participants’ demographics and background.

Demographics	n (%)	Background	n (%)
Current specialty		Patient species ^1^	
Veterinarians	52 (92.9)	Cats and/or dogs	51 (91.1)
Veterinary specialists	4 (7.1)	Small mammals, reptiles, exotics	5 (8.9)
Employment status		Others	5 (8.9)
Full time	43 (76.8)	Primary veterinary roles ^1^	
Part time	13 (23.2)	General practice	50 (89.3)
Earliest qualification		Emergency medicine	9 (16.1)
Less than 2 years ago	0 (0.0)	Others/refused to answer	7 (12.5)
2 < 5 years ago	4 (7.1)	Seniority	
5 < 8 years ago	9 (16.1)	Owner/partner	17 (30.4)
8 < 12 years ago	15 (26.8)	(Non-owner) key decision makers	6 (10.7)
12 < 16 years ago	7 (12.5)	Employee	33 (58.9)
16 < 20 years ago	7 (12.5)	Hours of sole charge per week ^2^	
20 < 25 years ago	6 (10.7)	Never the sole charge	10 (18.2)
More than 25 years	8 (14.3)	Sometimes sole charge	33 (60.0)
Gender ^2^		Always the sole charge	12 (21.8)
Male	7 (17.9)	Conventional working hours	
Female	30 (76.9)	Mostly between 08:00 and 20:00	44 (78.6)
Others/prefer not to say	2 (5.1)	Hours of work per week	42.2 ± 12.3

All data are represented in respondents (%) or mean ± SD, as appropriate. ^1^ Percentage exceeded 100% due to multiple responses. ^2^ Percentage based on valid sample. n—number of respondents in the sub-group. Missing values were excluded pairwise.

**Table 2 behavsci-13-00770-t002:** Binary logistics regression—mental health outcomes and ProQOL.

ProQOL Sub-Scales	Coef (B)	S.E.	Odds Ratio (Exp(B))	95% C.I.		Sig.
				Lower	Upper	
At risk of suicide						
BO score	−0.132	0.133	0.876	0.676	1.136	0.319
STS score	0.224	0.110	1.251	1.009	1.551	0.041
CS score	−0.165	0.097	0.848	0.701	1.026	0.089
Current suicidal ideation						
BO score	0.167	0.120	1.181	0.934	1.494	0.164
STS score	0.089	0.082	1.093	0.931	1.285	0.278
CS score	−0.039	0.089	0.962	0.807	1.145	0.660
Depression						
BO score	0.251	0.123	1.285	1.009	1.637	0.042
STS score	0.154	0.083	1.166	0.991	1.373	0.064
CS score	0.028	0.079	1.029	0.881	1.202	0.721
Anxiety						
BO score	0.199	0.110	1.221	0.985	1.513	0.069
STS score	0.130	0.076	1.139	0.981	1.321	0.088
CS score	0.039	0.074	1.039	0.900	1.201	0.600

ProQOL = Professional Quality of Lift scale; BO = burnout; STS = secondary traumatic stress; CS = compassion satisfaction.

## Data Availability

The data that support the findings of this study are available from the corresponding author upon reasonable request.

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
