# Peer review of "Predictors of Suicide Risk and Mental Health Outcomes among Hong Kong Veterinarians: A Cross-Sectional Study"

_behavsci, 2023, doi:10.3390/bs13090770_

Round 1

Reviewer 1 Report

Thank you for the opportunity to review this paper. This study investigates the impact of professional quality of life (ProQOL) on suicide risk, depression, and anxiety among Hong Kong veterinarians. The findings indicate high rates of these issues, comparable to or exceeding those in Anglophone and European countries. Burnout predicts depression, while secondary traumatic stress suggests potential suicide risk. Compassion satisfaction isn't conclusively linked to reduced symptoms. The study emphasizes the need for further exploration and improved mental health awareness and strategies within the veterinary profession. Overall, I believe the authors did a good job of designing the study and clearly presenting their findings.

I have the following suggestions/questions for the authors:

Regarding the sample selection, it would be helpful to provide more information about the distribution across these channels and any potential variations in response rates or demographics.

It could be beneficial to provide some additional context regarding the demographics, such as the gender distribution, years of experience, and types of veterinary practices the participants were engaged in. Is this information available?

Clarification on the handling of missing values could be included in the method section, particularly since some participants were excluded due to incomplete responses.

While the authors indicate that there are weak, moderate, and strong correlations between variable such as work hours and mental health scores, the discussion could elaborate on the practical implications of these correlations. For example, discuss why a weak or moderate correlation might still be significant in the context of veterinary practice. How might longer work hours contribute to mental health challenges? Is there a threshold beyond which work hours become particularly detrimental? Providing more context and interpretation here would be helpful.

The authors also indicate a correlation between works hours and GAD-7 scores. Provide further discussion about this. Why might longer work hours be associated with higher levels of anxiety?

The speculation on the veterinarian-client-patient dynamic and its impact on mental health is an interesting point. Consider exploring this further.

The conclusion section begins with an incomplete sentence. Is there some missing content here?

There are some minor grammatical/language errors. Please carefully copyedit to ensure the best possible presentation of your work.

Overall, the study presents valuable insights into the mental health landscape of veterinarians in Hong Kong. I believe the paper has the potential to make a valuable contribution to the literature, would be of interest to readers, and could be considered for publication after addressing the comments provided. I wish the authors all the best in their future work in this important field of study.

There are a few minor errors. Overall, the quality is adequate but could be improved for clarity.

Author Response

Dear Review 1 of our manuscript,

Thank you for your time commenting on our manuscript. We appreciate all your valuable comments.

Below, in point form, are our responses to your comment:

Regarding the sample selection, it would be helpful to provide more information about the distribution across these channels and any potential variations in response rates or demographics.

Reply: Thank you for your comment. Unfortunately, we did not collect the data as to how they were made aware of the survey, but we kept the dates to when the invitations and reminders were sent out. We have updated lines 91 to 96 and added lines 227-230 to elaborate address this and we added this as one of the study’s limitations in lines 472-475.

It could be beneficial to provide some additional context regarding the demographics, such as the gender distribution, years of experience, and types of veterinary practices the participants were engaged in. Is this information available?

Reply: Thank you for your comment. This information is available in Table 1 (lines 238-242) of the manuscript.

Clarification on the handling of missing values could be included in the method section, particularly since some participants were excluded due to incomplete responses.

Reply: Thank you for your comment. This information is available in lines 212-213. We have also added lines 213-215 to further clarifying it.

While the authors indicate that there are weak, moderate, and strong correlations between variable such as work hours and mental health scores, the discussion could elaborate on the practical implications of these correlations. For example, discuss why a weak or moderate correlation might still be significant in the context of veterinary practice. How might longer work hours contribute to mental health challenges? Is there a threshold beyond which work hours become particularly detrimental? Providing more context and interpretation here would be helpful.

Reply: Thank you for your comment. We have added lines 386-400 to discuss these issues.

The authors also indicate a correlation between works hours and GAD-7 scores. Provide further discussion about this. Why might longer work hours be associated with higher levels of anxiety?

Reply: Thank you for your comment. We have added lines 386-400 to discuss these issues.

The speculation on the veterinarian-client-patient dynamic and its impact on mental health is an interesting point. Consider exploring this further.

Reply: Thank you for your comment. We have added lines 408-416 to elaborate on this.

The conclusion section begins with an incomplete sentence. Is there some missing content here?

Reply: Thank you for the comment and we have added in the complete sentence.

There are some minor grammatical/language errors. Please carefully copyedit to ensure the best possible presentation of your work.

Reply: Thank you for the comment, and we have copyedited it for this revised version.

Overall, the study presents valuable insights into the mental health landscape of veterinarians in Hong Kong. I believe the paper has the potential to make a valuable contribution to the literature, would be of interest to readers, and could be considered for publication after addressing the comments provided. I wish the authors all the best in their future work in this important field of study.

Reply: Thank you very much for your kind words and we appreciate your valuable insights.

We hope the revision has clarified the manuscript and we look forward to hearing from you.

Regards,
Camille Chan

On behalf of all co-authors

Reviewer 2 Report

First of all, thank you very much for this opportunity to review the manuscript. Kindly find my comments below:

Abstract: If possible, please show some results of the logistic regression

I feel there is no need to use abbreviations such as BO, STS, etc in the abstract since they are mentioned only once.

Introduction

I also feel there is no need to use abbreviations in the manuscript text since they are not very commonly used.

For hypothesis (i), it may be difficult to test, since this is not a comparative study with other countries, and the measures used and the context of the study may be different. In hypothesis (ii), it is the same issue, since the data of HK medical professionals are not collected.

Use of the term "common mental illness" may be suitable in the introductory sentences, but I believe in the hypothesis it should be more definite, so may I suggest using depression and anxiety symptoms.

Methods

Line 72, the authors stated, "this study design" please replace with the "cross-sectional study design" to clarify the pronoun.

Line 74, it may be unsuitable to use the term "impact" for a cross-sectional study as impact is usually measured across time. Perhaps use the word association or relationship

Line 76-81, In the study setting, please mention universal recruitment was used. Also, since a few settings were used to recruit, and some veterinarians belong to two or more settings, is there any possibility that they respond twice? Were there any measures to detect/prevent this from happening?

Sorry I am ignorant about the practice of veterinary in HK. In the abstract, it was mentioned that the vets are "registered and practicing", but only practicing vets are stated as an inclusion criteria in the methods section. Could the authors clarify this, did they want to recruit registered and practicing vets? Are there vets who do not register but still practice?

line 86: e-screening was used. May I know what were the screening questions? And how were participants screened out? Kindly provide more details.

Line 89 : It may be useful to state English is a second and commonly used language in HK SAR

Could the authors provide the approximate time to complete the 140 questions, were there any potential to induce fatigue?

The response options and scoring for each option was not provided for PRQoL and SBQ-R. Perhaps the authors could add this for clarity.

What are the cutoffs used for all instruments, if applicable? Could the authors provide a reference for using the proposed cut offs?

line 146 : Reminders were sent, were they done systematically? If yes, how many reminders were sent, and in what interval?

Study size, is the study adequately powered? Please either show a sample size calculation or a reference for the rule-of-thumb used.

Line 162: It is not clear if the authors are conducting a simple logistic regression or a multiple logistic regression? If simple regression, why not use a multiple logistic regression?

ll. 167-168: "The strength and direction of linear relationship between hours of work per week and SBQ-R, PHQ-9, and GAD-7 scores were explored using correlation." Where are the results?

line 197: Without providing the cut offs in the methods section, it is difficult to see how you derived 22%.

line 228: I don't think it is appropriate to use the term "condition" for depression and anxiety as they were only screened, not diagnosed. Perhaps it is better for the authors to state depression and anxiety symptoms, or screened positive for depression and anxiety symptoms.

For significance, please provide up to three decimals. Also, kindly amend the use of p = 0.00.

Line 243: Rather than use the word contribution, which denotes causality, please use the word association

Table 2: Please indicate whether it is a multiple logistic regression

The discussion looks good to me

Line 380 appears incomplete to me.

Please state also in the limitations that 6.6% response rate is very low and susceptible to bias.

Thank you for the consideration of these comments.

Author Response

Dear Review 2 of our manuscript,

Thank you for your time commenting on our manuscript. We appreciate all your valuable comments.

Below, in point form, are our responses to your comment:

Abstract:

If possible, please show some results of the logistic regression

Reply: Thank you. Results from the logistic regression was reported in lines 17-19, and we have added a short phrase before it to make it more explicit.

I feel there is no need to use abbreviations such as BO, STS, etc in the abstract since they are mentioned only once.

Reply: Thank you. We agree that abbreviations are not needed in the abstract and have removed them.

Introduction

I also feel there is no need to use abbreviations in the manuscript text since they are not very commonly used.

Reply: Thank you for the comment. We removed the abbreviations of BO, STS, and CS throughout the manuscript.

For hypothesis (i), it may be difficult to test, since this is not a comparative study with other countries, and the measures used and the context of the study may be different. In hypothesis (ii), it is the same issue, since the data of HK medical professionals are not collected.

Use of the term "common mental illness" may be suitable in the introductory sentences, but I believe in the hypothesis it should be more definite, so may I suggest using depression and anxiety symptoms.

Reply: Thank you for your comment. We revised the hypotheses and we discarded the ‘comparison’ wordings in the hypotheses and only focused on the data that directly collected from our study, but remain the comparison of findings only in the discussion section.

Methods

Line 72, the authors stated, "this study design" please replace with the "cross-sectional study design" to clarify the pronoun.

Reply: Thank you and we agree. In fact, we rephrased the sentences (lines 80-87) to improve the sentence and paragraph structure.

Line 74, it may be unsuitable to use the term "impact" for a cross-sectional study as impact is usually measured across time. Perhaps use the word association or relationship

Reply: Thank you and we agree. We rephrased the sentences (lines 80-87) to improve the sentence and paragraph structure.

Line 76-81, In the study setting, please mention universal recruitment was used. Also, since a few settings were used to recruit, and some veterinarians belong to two or more settings, is there any possibility that they respond twice? Were there any measures to detect/prevent this from happening?

Reply: Thank you for your comment. we have added lines 91-99 that explained the potential reach of these invitations to veterinarians in Hong Kong, as well as lines 468-469 and 472-475 to address the universal recruitment and hinted the unlikelihood of duplicated response due to the length of the questionnaire.

Sorry I am ignorant about the practice of veterinary in HK. In the abstract, it was mentioned that the vets are "registered and practicing", but only practicing vets are stated as an inclusion criteria in the methods section. Could the authors clarify this, did they want to recruit registered and practicing vets? Are there vets who do not register but still practice?

Reply: Thank you for the remark and it is our pleasure to learn from you and make this manuscript clearer for everyone. In Hong Kong, veterinarians cannot practice unless they have registered with the Veterinary Surgeon Board of Hong Kong, therefore there are chances where veterinarians are registered but wasn’t working. But there is no chance that a veterinarian who works is not registered in Hong Kong, unless they have broken the law, which, to our knowledge, is extremely rare.

I have modified line 101-102 to improve clarity.

line 86: e-screening was used. May I know what were the screening questions? And how were participants screened out? Kindly provide more details.

Reply: Thank you for your comment. We have added in lines 135-136 that guides readers to the supplementary material, where the screening questions and main questionnaires are attached.

Line 89 : It may be useful to state English is a second and commonly used language in HK SAR

Reply: Thank you for your comment. We have updated lines 108-112 to address this.

Could the authors provide the approximate time to complete the 140 questions, were there any potential to induce fatigue?

Reply: Thank you for your comment. We have added lines 468-469 to address this as our limitation.

The response options and scoring for each option was not provided for PRQoL and SBQ-R. Perhaps the authors could add this for clarity.

Reply: Thank you for your comment. We have added in lines 135-136 that guides readers to the supplementary material, where the screening questions and main questionnaires are attached.

What are the cutoffs used for all instruments, if applicable? Could the authors provide a reference for using the proposed cut offs?

Reply: Thank you for your comment. The cut-off score for ProQOL are reported in lines 144-146, for PHQ-9 in lines 164-166, and for GAD-7 in lines 172-174. We have added in line 154 the cut-off scores for SBQ-R.

line 146 : Reminders were sent, were they done systematically? If yes, how many reminders were sent, and in what interval?

Reply: Thank you for your comment. We added that reminders were sent about two months after the first invitations were sent through different channels in lines 97 to 99.

Study size, is the study adequately powered? Please either show a sample size calculation or a reference for the rule-of-thumb used.

Reply: Thank you for the invitation and since the studied population is relatively small and hence, we adopted the rule-of-thumb for sample estimation instead and is reported in lines 187-191.

Line 162: It is not clear if the authors are conducting a simple logistic regression or a multiple logistic regression? If simple regression, why not use a multiple logistic regression?

Reply: Thank you for your comment. We used binary logistic regression to measure the relationship between the predictors and dependent variables. We agree the results would be more robust if multiple logistic regression was used, but we were restricted by a sample size that cannot reach a minimum of 20 observations per independent variable. We have added lines 202-208 to justify our use of logistic regression.

  1. 167-168: "The strength and direction of linear relationship between hours of work per week and SBQ-R, PHQ-9, and GAD-7 scores were explored using correlation." Where are the results?

Reply: Thank you for your comment. The results are reported in lines 231-236.

line 197: Without providing the cut offs in the methods section, it is difficult to see how you derived 22%.

Reply: Thank you for your comment. We have added in line 154 the cut-off scores for SBQ-R.

line 228: I don't think it is appropriate to use the term "condition" for depression and anxiety as they were only screened, not diagnosed. Perhaps it is better for the authors to state depression and anxiety symptoms, or screened positive for depression and anxiety symptoms.

Reply: Thank you for your comment. We used “symptoms” instead of conditions in the revised version of the manuscript.

For significance, please provide up to three decimals. Also, kindly amend the use of p = 0.00.

Reply: Thank you for your comment. We have changed the word from including lines 234-236 and 277-280.

Line 243: Rather than use the word contribution, which denotes causality, please use the word association

Reply: Thank you for your comment. We have changed the word from “contribution” to “association”.

Table 2: Please indicate whether it is a multiple logistic regression

Reply: Thank you for your comment. We used simple logistic regression which to measure the binary outcomes regarding relationship between the predictors and dependent variables (i.e. binary logistic regression). We have modified its caption to “binary logistic regression” for clarity.

The discussion looks good to me

Reply: Thank you.

Line 380 appears incomplete to me.

Reply: That is correct and we apologise for our carelessness. We have added in the complete sentence.

Please state also in the limitations that 6.6% response rate is very low and susceptible to bias.

Reply: Thank you. We agree that it is susceptible to bias and have added it in line 468-469.

We hope the revision has clarified the manuscript and we look forward to hearing from you.

Regards,
Camille Chan

On behalf of all co-authors

Round 2

Reviewer 2 Report

Thank you, I find that the authors have addressed all my questions satisfactorily, and there are no major comments from me. Just some minor comments:

I feel that the conclusions should only state a summary of the study, rather than provide new information. If the authors could reword the Conclusions, and move the implications to before the Conclusions section, it would be great.

Some spelling errors, e.g., line 429 "sued" should be "used", line 333 "pour" should be "poor", the sentence in lines 320-322 appears to be hanging.

Thank you
